# Identification of drugs associated with reduced severity of COVID-19 – a case-control study in a large population

Ariel Israel[1]*, Alejandro A Schäffer[2], Assi Cicurel[1,3], Kuoyuan Cheng[2], Sanju Sinha[2], Eyal Schiff[4], Ilan Feldhamer[1], Ameer Tal[1], Gil Lavie[1,5]†, Eytan Ruppin[2]†*

[1]Division of Planning and Strategy, Clalit Health Services, Tel Aviv, Israel; [2]Cancer Data Science Laboratory, National Cancer Institute, National Institutes of Health, Bethesda, United States; [3]Clalit Health Services, Southern District and Faculty of Health Sciences, Ben-Gurion University of the Negev, Beer-Sheva, Israel; [4]Sheba Medical Center, Tel-Aviv University, Ramat Gan, Israel; [5]Ruth and Bruce Rappaport Faculty of Medicine, Technion – Israel Institute of Technology, Haifa, Israel

**\*For correspondence:**
dr.ariel.israel@gmail.com (AI);
eytan.ruppin@nih.gov (ER)

†These authors contributed equally to this work

**Competing interests:** The authors declare that no competing interests exist.

## Abstract

**Background:** Until coronavirus disease 2019 (COVID-19) drugs specifically developed to treat COVID-19 become more widely accessible, it is crucial to identify whether existing medications have a protective effect against severe disease. Toward this objective, we conducted a large population study in Clalit Health Services (CHS), the largest healthcare provider in Israel, insuring over 4.7 million members.

**Methods:** Two case-control matched cohorts were assembled to assess which medications, acquired in the last month, decreased the risk of COVID-19 hospitalization. Case patients were adults aged 18 to 95 hospitalized for COVID-19. In the first cohort, five control patients, from the general population, were matched to each case (n=6202); in the second cohort, two non-hospitalized SARS-CoV-2 positive control patients were matched to each case (n=6919). The outcome measures for a medication were: odds ratio (OR) for hospitalization, 95% confidence interval (CI), and the p-value, using Fisher's exact test. False discovery rate was used to adjust for multiple testing.

**Results:** Medications associated with most significantly reduced odds for COVID-19 hospitalization include: ubiquinone (OR=0.185, 95% CI [0.058 to 0.458], p<0.001), ezetimibe (OR=0.488, 95% CI [0.377 to 0.622], p<0.001), rosuvastatin (OR=0.673, 95% CI [0.596 to 0.758], p<0.001), flecainide (OR=0.301, 95% CI [0.118 to 0.641], p<0.001), and vitamin D (OR=0.869, 95% CI [0.792 to 0.954], p<0.003). Remarkably, acquisition of artificial tears, eye care wipes, and several ophthalmological products were also associated with decreased risk for hospitalization.

**Conclusions:** Ubiquinone, ezetimibe, and rosuvastatin, all related to the cholesterol synthesis pathway were associated with reduced hospitalization risk. These findings point to a promising protective effect which should be further investigated in controlled, prospective studies.

**Funding:** This research was supported in part by the Intramural Research Program of the National Institutes of Health, NCI.

## Introduction

SARS-CoV-2 is a new single-stranded RNA virus, which was first identified in December 2019, and has rapidly spread into a global pandemic of primarily respiratory illness designated as coronavirus disease 2019 (COVID-19). This disease is associated with significant mortality, particularly among

elderly or overweight individuals, raising considerable concerns for public health. Until a vaccine or specifically designed therapies are available, it is urgent to identify whether existing medications have protective effects against COVID-19 complications using available real-world data. With this aim, we performed a case-control study on electronic health records (EHRs) from Clalit Health Services (CHS), the largest healthcare provider in Israel.

## Materials and methods

### Participants and data sources

We collected data from the CHS data warehouse on adult patients aged 18 to 95 years, who tested positive for SARS-CoV-2 from the beginning of the pandemic through November 30, 2020, and were admitted for hospitalization through December 31, 2020. Each patient was assigned an index date, which is the first date at which a positive RT-PCR test for SARS-CoV-2 was collected for the patient. Patients' demographic characteristics were extracted, along with existing comorbidities, clinical characteristics including body mass index (BMI), and estimated glomerular filtration rate (eGFR) at the baseline, defined as of February 2020. In addition, the list of drugs or products acquired by each patient in CHS pharmacies was collected for the month preceding the index date, defined as the 35 days prior to this date.

Reliable identification of medications procured for a given month is enabled by the fact that in CHS, distinct prescriptions are issued for each calendar month. When medications are provided in advance for multiple months, the date at which the prescription for each month of treatment begins is recorded.

This study has been approved by the CHS Institutional Review Board (IRB) with a waiver of informed consent, approval number: COM-0046–20. Patient data that could identify participants were removed prior to the statistical analyses in accordance with the protocol approved by the CHS IRB.

### Software

Patients' data were extracted and processed from CHS data warehouse using programs developed in-house in Python and SQL.

### Case-control design and matching

Hospitalized COVID-19 patients were assigned to two distinct case-control cohorts, which differ in the way control individuals were selected. In *cohort 1*, control patients were chosen among the general population of CHS members. Since controls can be selected from among millions of individuals, five controls were selected to match each case (5:1), with comprehensively matched baseline attributes, including age, sex, BMI category, socio-economic and smoking status, chronic kidney disease (CKD) stage for patients with renal impairment, and main comorbidities diagnoses (hypertension, diabetes, CKD, congestive heart failure [CHF], chronic obstructive pulmonary disease [COPD], malignancy, ischemic heart disease). For the matching procedure, patients with undocumented BMI were considered as having a normal BMI, unless an obesity diagnosis was present. Each control was assigned the same index date as the matched case, provided that the patient was still alive and a member of CHS at this date. EHR data were collected for controls using the same procedure described for cases. *Cohort 1* is designed to identify drugs that affect the overall risk for hospitalization for COVID-19, where the effect could combine a decreased risk of detectable infection, and a decreased risk for hospitalization once infected.

In *cohort 2*, control patients were chosen among patients who had a positive test for SARS-CoV-2 but had not been hospitalized as of December 31, 2020. Given the smaller size of the pool from which controls can be drawn, only two controls were matched for each case patient. Attributes that were matched were the age, sex, smoking status, Adjusted Clinical Groups (ACG) measure of comorbidity (*Shadmi et al., 2011*) and presence/absence of an obesity diagnosis. The index date taken was the date of the first positive SARS-CoV-2 PCR test both for cases and for controls. *Cohort 2* is more specifically suited to identify drugs that are associated with a decreased risk for COVID-19 hospitalization in patients who had a proven infection with the virus. In both cohorts, there were a minority of case individuals for which enough matching controls could not be found; these cases

were not included in their respective cohorts. Patients who were pregnant since February 2020 were also excluded.

## Outcome measures

In each cohort, and for each medication anatomical therapeutic chemical (ATC) class, the odds ratio (OR) for hospitalization was computed, comparing the number of patients who acquired a medication belonging to the class in the 35 days preceding the index date, in the case and the control groups.

## Statistical analysis

OR for hospitalization for drugs acquired in the case versus control groups and statistical significance were assessed by Fisher's exact test. Correction for multiple testing was performed using the Benjamini-Hochberg procedure (*Benjamini and Hochberg, 1995*), which gives an estimation of the false discovery rate (FDR) in the list. To assess the effects of being in one of two high-risk subgroups, Ultra-Orthodox Jews and Arabs, we used multivariable conditional logistic regression analyses performed in each of the cohorts. In each cohort, we modelized the OR for hospitalization, using subgroup membership and purchased medications as explanatory factors.

To assess for possible associations between the protective effect of a medication and BMI, we partition the matched subjects into four BMI ranges: <25, 25 to 30, 30 to 35, >35. Then we redid our association analyses in each range.

Statistical analyses were performed in R statistical software version 3.6 (R Foundation for statistical computing).

## Role of the funding source

The funder of the study had no role in study design, data collection, data analysis, data interpretation, or writing of the report. AI, IF, and AT had full access to all the data in the study and had final responsibility for the decision to submit for publication.

## Results

Through December 31, 2020, 10,295 adult patients between the ages of 18 and 95 had a recorded COVID-19 related hospitalization in the CHS database. The matching procedure was able to identify control individuals from the general population in ratio 5:1 for 6530 patients in the first cohort, and control patients in ratio 2:1 for 6953 SARS-CoV-2 positive individuals in the second cohort. The characteristics of the matched populations are shown in *Table 1*.

In each of the two cohorts, we counted the number of patients from each group who acquired drugs and other medical products from each ATC class and computed the OR and p-values using Fisher's exact test. The distribution of OR for drugs for which the p-value was statistically significant (p<0.05) is shown in *Figure 1*. The OR for most drugs are neutral or associated with an increased risk of COVID-19 hospitalization. Only a small number of items are associated with decreased risk: 1.15% in cohort 1 and 1.75% in cohort 2.

*Table 2* presents the list of drugs and products that were found to be negatively associated with COVID-19 hospitalization in a statistically significant manner in cohort 1 (A) and in cohort 2 (B). We display items for which the p-value is below 0.05, and for which the FDR is less than 0.20, meaning that at least 80% of the items in the displayed list are expected to be true positives. Items are sorted in decreasing order of significance.

The top ranked medications by significance in *cohort 1* were rosuvastatin (OR=0.673, 95% confidence interval [CI] 0.596 to 0.758), ezetimibe (OR=0.488, CI 0.377 to 0.622), and ubiquinone (OR=0.181, CI 0.065 to 0.403); these same three medications were also in the top five by significance of *cohort 2*: rosuvastatin (OR=0.732, CI 0.643 to 0.83), ezetimibe (OR=0.602; CI 0.471 to 0.764), and ubiquinone (OR=0.181, CI 0.065 to 0.403). It is remarkable that these three drugs act on the cholesterol and ubiquinone synthesis pathways, which both stem from the mevalonate pathway (*Buhaescu and Izzedine, 2007*); the intermediate product at the branch point is farnesyl polyphosphate (FPP) (*Figure 2*). Rosuvastatin and other statins specifically inhibit he enzyme HMG-CoA reductase. Ubiquinone is a food supplement available over the counter, which is often recommended to patients prone to muscular pain and receiving a statin treatment (*Qu et al., 2018*).

**Table 1.** Demographics and clinical characteristics of the two matched cohorts of patients (hospitalized versus non-hospitalized).

| | Cohort 1 | | Cohort 2 | |
| --- | --- | --- | --- | --- |
| | COVID-19 hospitalized (cases) | Not hospitalized (controls) | COVID-19 hospitalized (cases) | Not hospitalized (controls) |
| n | 6530 | 32,650 | 6953 | 13,906 |
| Age (mean, SD) | 64.6 (16.1) | 64.8 (15.8) | 65.7 (16.0) | 65.7 (15.8) |
| Sex, female (%) | 3259 (49.9) | 16,295 (49.9) | 3381 (48.6) | 6762 (48.6) |
| Hospitalization severity (n, %) | | | | |
| Mild condition | 3008 (46.1) | | 2676 (38.5) | |
| Serious condition | 851 (13.0) | | 1043 (15.0) | |
| Severe condition | 1621 (24.8) | | 1903 (27.4) | |
| Deceased | 1050 (16.1) | | 1331 (19.1) | |
| Smoking status (%) | | | | |
| Never smoker | 5012 (76.8) | 24,218 (74.2) | 5156 (74.2) | 10,312 (74.2) |
| Past smoker | 1115 (17.1) | 5808 (17.8) | 1338 (19.2) | 2676 (19.2) |
| Current smoker | 403 (6.2) | 2624 (8.0) | 459 (6.6) | 918 (6.6) |
| Nb visits at primary doctor in last year (mean, SD) | 8.1 (7.9) | 7.9 (7.3) | 8.2 (8.4) | 7.5 (7.1) |
| Comorbidity (%) | | | | |
| Arrhythmia | 887 (13.6) | 4242 (13.0) | 1278 (18.4) | 2221 (16.0) |
| Asthma | 527 (8.1) | 2941 (9.0) | 650 (9.3) | 1376 (9.9) |
| Congestive heart failure (CHF) | 228 (3.5) | 1140 (3.5) | 784 (11.3) | 851 (6.1) |
| Chronic obstructive pulmonary disease (COPD) | 148 (2.3) | 740 (2.3) | 603 (8.7) | 776 (5.6) |
| Diabetes | 2976 (45.6) | 14,880 (45.6) | 3425 (49.3) | 5549 (39.9) |
| Hypertension | 3850 (59.0) | 19,062 (58.4) | 4396 (63.2) | 8102 (58.3) |
| Ischemic heart disease (IHD) | 1464 (22.4) | 7320 (22.4) | 1838 (26.4) | 3113 (22.4) |
| Malignancy | 1087 (16.6) | 5435 (16.6) | 1280 (18.4) | 2766 (19.9) |
| Chronic kidney disease (CKD) | 102 (1.6) | 510 (1.6) | 1086 (15.6) | 1117 (8.0) |
| Obesity (documented diagnosis) | 3761 (57.6) | 17,837 (54.6) | 3975 (57.2) | 7950 (57.2) |
| Body mass index (BMI) (mean, SD) | 28.7 (5.7) | 28.6 (6.5) | 29.1 (6.3) | 28.5 (5.7) |
| BMI group (%) | | | | |
| <18.5 | 17 (0.3) | 85 (0.3) | 51 (0.7) | 93 (0.7) |
| 18.5 to 25 | 1070 (16.4) | 5350 (16.4) | 1244 (17.9) | 2471 (17.8) |
| 25 to 30 | 2295 (35.1) | 11,475 (35.1) | 2264 (32.6) | 4870 (35.0) |
| 30 to 35 | 2053 (31.4) | 10,265 (31.4) | 2005 (28.8) | 4267 (30.7) |
| 35 to 40 | 761 (11.7) | 3805 (11.7) | 886 (12.7) | 1562 (11.2) |
| >40 | 334 (5.1) | 1670 (5.1) | 503 (7.2) | 643 (4.6) |
| Glomerular filtration rate (GFR) (mean, SD) | 85.7 (21.6) | 85.8 (20.3) | 78.7 (28.2) | 83.4 (22.4) |
| Chronic kidney disease (CKD) staging (n, %) | | | | |
| G1 | 3047 (46.7) | 15,145 (46.4) | 2837 (40.8) | 6090 (43.8) |
| G2 | 2679 (41.0) | 13,747 (42.1) | 2535 (36.5) | 5722 (41.1) |
| G3a | 558 (8.5) | 2817 (8.6) | 689 (9.9) | 1257 (9.0) |
| G3b | 203 (3.1) | 836 (2.6) | 391 (5.6) | 571 (4.1) |
| G4 | 41 (0.6) | 89 (0.3) | 186 (2.7) | 160 (1.2) |
| G5 | | | 63 (0.9) | 28 (0.2) |
| Dialysis | 2 (0.0) | 16 (0.0) | 252 (3.6) | 78 (0.6) |

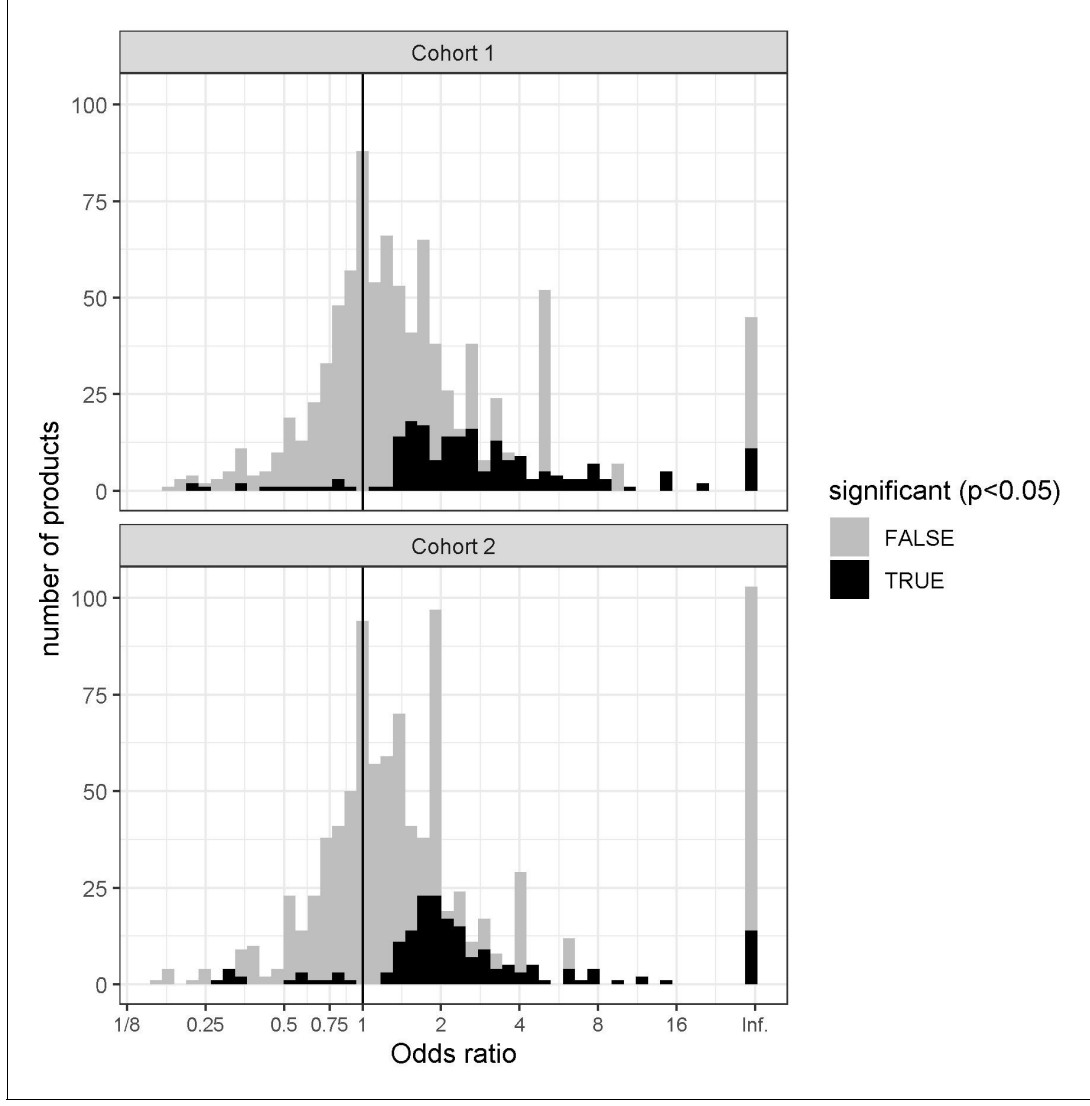

**Figure 1.** Histogram showing the distribution of the odd ratios (OR) of medication use with the outcome in cohorts 1 and 2. The overwhelming majority of medications are associated with neutral effect (gray) or increased risk for hospitalization (black, OR>1), only a few are associated with significantly decreased risk (black, OR<1).

Risedronate, which also acts on this pathway, and is commonly used to prevent osteoporosis, by blocking the enzyme FPP synthase, is also identified by both cohorts, and is ranked 4th by significance in *cohort 2* (OR=0.567; CI 0.400 to 0.789), and 13th in *cohort 1* (OR=0.705; CI 0.522 to 0.935).

Other medications that fulfilled the stringent criteria of being identified by both cohorts with an FDR of 80% include the pneumococcal conjugate vaccine (OR=0.476, CI 0.288 to 0.746 in *cohort 1*; 0.602, CI 0.245 to 0.685 in *cohort 2*), magnesium citrate (OR=0.652, CI 0.433 to 0.952 in *cohort 1*; 0.609, CI 0.399 to 0.908 in *cohort 2*), vitamin D (OR=0.898, CI 0.821 to 0.980 in *cohort 1*; 0.869, CI 0.792 to 0.954 in *cohort 2*), flecainide (OR=0.301, CI 0.118 to 0.641 in *cohort 1*; 0.325, CI 0.123 to 0.729 in *cohort 2*), escitalopram (OR=0.824, CI 0.708 to 0.955 in *cohort 1*; 0.766, CI 0.654 to 0.894 in *cohort 2*), cilazapril (OR=0.554, CI 0.315 to 0.918 in *cohort 1*; 0.468, CI 0.247 to 0.831 in *cohort 2*), ramipril combined with hydrochlorothiazide (OR=0.734, CI 0.603 to 0.887 in *cohort 1*; 0.702, CI 0.565 to 0.869 in *cohort 2*), and sitagliptin combined with metformin (OR=0.802, CI 0.696 to 0.922 in *cohort 1*; 0.826, CI 0.704 to 0.967 in *cohort 2*). Sitagliptin alone is also significant in *cohort 1* (OR=0.658, CI 0.443 to 0.950).

In addition, we observe interesting patterns in *cohort 2,* which is designed to identify drugs associated with decreased hospitalization risk in SARS-CoV-2 positive patients: several vitamin or mineral

**Table 2.** Most significant associations for medications acquired in the 35 days preceding the index date in two matched cohorts.

| ATC code and class | Use in case | Use in contr. | Case % | Contr. % | Odds ratio (95% conf. int.) | p-Value | FDR |
|---|---|---|---|---|---|---|---|
| (A) Cohort 1 (N = 6530 hospitalization cases, N=32,650 controls taken from the general population) | | | | | | | |
| C10AA07 Rosuvastatin | 328 | 2380 | 5.02 | 7.29 | 0.673 (0.596 to 0.758) | <0.0001 | <0.001 |
| C10AX09 Ezetimibe | 73 | 740 | 1.12 | 2.27 | 0.488 (0.377 to 0.622) | <0.0001 | <0.001 |
| A16AX30 Ubiquinone (CoQ-10) | 6 | 165 | 0.09 | 0.51 | 0.181 (0.065 to 0.403) | <0.0001 | <0.001 |
| C01BC04 Flecainide | 7 | 116 | 0.11 | 0.36 | 0.301 (0.118 to 0.641) | 0.00039 | 0.005 |
| J07AL02 Pneumococcus vaccine conjugate | 21 | 220 | 0.32 | 0.67 | 0.476 (0.288 to 0.746) | 0.00049 | 0.006 |
| C09BA05 Ramipril-hydrochlorothiazide | 127 | 859 | 1.95 | 2.63 | 0.734 (0.603 to 0.887) | 0.00099 | 0.011 |
| A10BD07 Sitagliptin-metformin | 243 | 1501 | 3.72 | 4.60 | 0.802 (0.696 to 0.922) | 0.00159 | 0.017 |
| C10AA03 Pravastatin | 52 | 385 | 0.80 | 1.18 | 0.673 (0.493 to 0.902) | 0.00659 | 0.060 |
| N06AB10 Escitalopram | 216 | 1302 | 3.31 | 3.99 | 0.824 (0.708 to 0.955) | 0.00930 | 0.078 |
| M01AC01 Piroxicam | 24 | 205 | 0.37 | 0.63 | 0.584 (0.365 to 0.894) | 0.00980 | 0.082 |
| C09CA06 Candesartan | 65 | 451 | 1.00 | 1.38 | 0.718 (0.544 to 0.934) | 0.01237 | 0.100 |
| M05BA07 Risedronic acid | 56 | 396 | 0.86 | 1.21 | 0.705 (0.522 to 0.935) | 0.01319 | 0.103 |
| G04CB02 Dutasteride | 30 | 240 | 0.46 | 0.74 | 0.623 (0.411 to 0.914) | 0.01367 | 0.105 |
| A11CC05 Cholecalciferol | 660 | 3634 | 10.11 | 11.13 | 0.898 (0.821 to 0.980) | 0.01600 | 0.119 |
| C09AA08 Cilazapril | 17 | 153 | 0.26 | 0.47 | 0.554 (0.315 to 0.918) | 0.01743 | 0.124 |
| G04BE08 Tadalafil | 29 | 229 | 0.44 | 0.70 | 0.632 (0.413 to 0.933) | 0.01862 | 0.132 |
| S01ED61 Timolol-travoprost | 8 | 90 | 0.12 | 0.28 | 0.444 (0.186 to 0.913) | 0.02068 | 0.142 |
| A10BH01 Sitagliptin | 33 | 250 | 0.51 | 0.77 | 0.658 (0.443 to 0.950) | 0.02461 | 0.162 |
| J07BB02 Influenza vaccine inac | 392 | 2205 | 6.00 | 6.75 | 0.882 (0.787 to 0.986) | 0.02548 | 0.165 |
| N06DX02 Ginkgo folium | 2 | 42 | 0.03 | 0.13 | 0.238 (0.028 to 0.915) | 0.02552 | 0.165 |
| A12CC04 Magnesium citrate | 31 | 237 | 0.48 | 0.73 | 0.652 (0.433 to 0.952) | 0.02597 | 0.166 |
| A10BK01 Dapagliflozin | 35 | 255 | 0.54 | 0.78 | 0.685 (0.466 to 0.978) | 0.03283 | 0.193 |
| (B) Cohort 2 (N = 6953 hospitalization cases, N=13,906 controls taken from patients SARS-CoV-2 positive) | | | | | | | |
| C10AA07 Rosuvastatin | 354 | 950 | 5.09 | 6.83 | 0.732 (0.643 to 0.831) | <0.0001 | 0.000 |
| C10AX09 Ezetimibe | 92 | 303 | 1.32 | 2.18 | 0.602 (0.471 to 0.764) | 0.00001 | 0.000 |
| J07AL02 Pneumococcus vaccine conjugate | 20 | 95 | 0.29 | 0.68 | 0.419 (0.245 to 0.685) | 0.00021 | 0.003 |

*Table 2 continued on next page*

*Table 2 continued*

| ATC code and class | Use in case | Use in contr. | Case % | Contr. % | Odds ratio (95% conf. int.) | p-Value | FDR |
|---|---|---|---|---|---|---|---|
| M05BA07 Risedronic acid | 47 | 165 | 0.68 | 1.19 | 0.567 (0.400 to 0.789) | 0.00042 | 0.005 |
| A16AX30 Ubiquinone (CoQ-10) | 9 | 56 | 0.13 | 0.40 | 0.321 (0.139 to 0.653) | 0.00052 | 0.006 |
| N06AB10 Escitalopram | 236 | 610 | 3.39 | 4.39 | 0.766 (0.654 to 0.894) | 0.00061 | 0.007 |
| C09BA05 Ramipril-hydrochlorothiazide | 121 | 342 | 1.74 | 2.46 | 0.702 (0.565 to 0.869) | 0.00082 | 0.009 |
| C01BC04 Flecainide | 7 | 43 | 0.10 | 0.31 | 0.325 (0.123 to 0.729) | 0.00253 | 0.023 |
| S01XA40 Hydroxypropyl-methylcellulose (tears) | 67 | 203 | 0.96 | 1.46 | 0.657 (0.490 to 0.871) | 0.00273 | 0.025 |
| A11CC05 Cholecalciferol | 737 | 1669 | 10.60 | 12.00 | 0.869 (0.792 to 0.954) | 0.00280 | 0.025 |
| B01AE07 Dabigatran etexilate | 37 | 124 | 0.53 | 0.89 | 0.595 (0.400 to 0.866) | 0.00543 | 0.042 |
| C09AA08 Cilazapril | 15 | 64 | 0.22 | 0.46 | 0.468 (0.247 to 0.831) | 0.00579 | 0.044 |
| N02CC04 Rizatriptan | 1 | 17 | 0.01 | 0.12 | 0.118 (0.003 to 0.750) | 0.01065 | 0.075 |
| A12CC04 Magnesium citrate | 33 | 108 | 0.48 | 0.78 | 0.609 (0.399 to 0.908) | 0.01191 | 0.080 |
| S01KA01 Hyaluronic acid (artificial tears) | 5 | 31 | 0.07 | 0.22 | 0.322 (0.098 to 0.836) | 0.01249 | 0.083 |
| C09DB01 Valsartan-amlodipine | 227 | 549 | 3.27 | 3.95 | 0.821 (0.698 to 0.963) | 0.01445 | 0.094 |
| A10BD07 Sitagliptin-metformin | 233 | 560 | 3.35 | 4.03 | 0.826 (0.704 to 0.967) | 0.01721 | 0.108 |
| B03BA51 Vit.B12 combinations | 31 | 100 | 0.45 | 0.72 | 0.618 (0.399 to 0.934) | 0.01979 | 0.119 |
| G03CA03 Estradiol | 18 | 67 | 0.26 | 0.48 | 0.536 (0.300 to 0.914) | 0.02047 | 0.122 |
| C09DA01 Losartan-hydrochlorothiazide | 124 | 315 | 1.78 | 2.27 | 0.783 (0.630 to 0.969) | 0.02424 | 0.140 |
| S01ED01 Timolol | 20 | 70 | 0.29 | 0.50 | 0.570 (0.328 to 0.949) | 0.02492 | 0.143 |
| G04BD12 Mirabegron | 22 | 74 | 0.32 | 0.53 | 0.593 (0.351 to 0.967) | 0.02998 | 0.163 |
| S01XA02 Retinol (eye ointment) | 3 | 21 | 0.04 | 0.15 | 0.285 (0.054 to 0.956) | 0.03015 | 0.163 |
| Z01CE01 Eye care wipes | 3 | 21 | 0.04 | 0.15 | 0.285 (0.054 to 0.956) | 0.03015 | 0.163 |
| N06AX12 Bupropion | 6 | 30 | 0.09 | 0.22 | 0.399 (0.136 to 0.976) | 0.03385 | 0.177 |
| N06BA04 Methylphenidate | 8 | 36 | 0.12 | 0.26 | 0.444 (0.178 to 0.972) | 0.03656 | 0.186 |
| A12AX05 Calcium-zinc CD | 0 | 10 | 0.00 | 0.07 | 0.000 (0.000 to 0.892) | 0.03696 | 0.186 |
| A11JC02 Multivitamins for ocular use | 25 | 81 | 0.36 | 0.58 | 0.616 (0.376 to 0.976) | 0.03827 | 0.191 |

Numbers are of patients from the group who have acquired a medication from the class in the last month before the index date.

p-Values are calculated according to Fisher's exact test. Medications are sorted by increasing order of p-values.

OR: odds ratio; [95% CI]: 95% confidence interval; FDR: false discovery rate calculated according to Benjamini-Hochberg (BH) procedure.

Shown in this table are anatomical therapeutic chemical (ATC) classes for which the p-value is less than 0.05, and for which the FDR is less than 0.20 (about 80% of entries are expected to be true positive).

supplementation items appear to have a protective effect, in addition to vitamin D and magnesium citrate, which were identified by both cohorts: vitamin B12 combinations (OR=0.618, CI 0.399 to 0.934), multivitamins for ocular use (OR=0.616, CI 0.376 to 0.976), and calcium-zinc combinations (OR=0.000, CI 0.000 to 0.892).

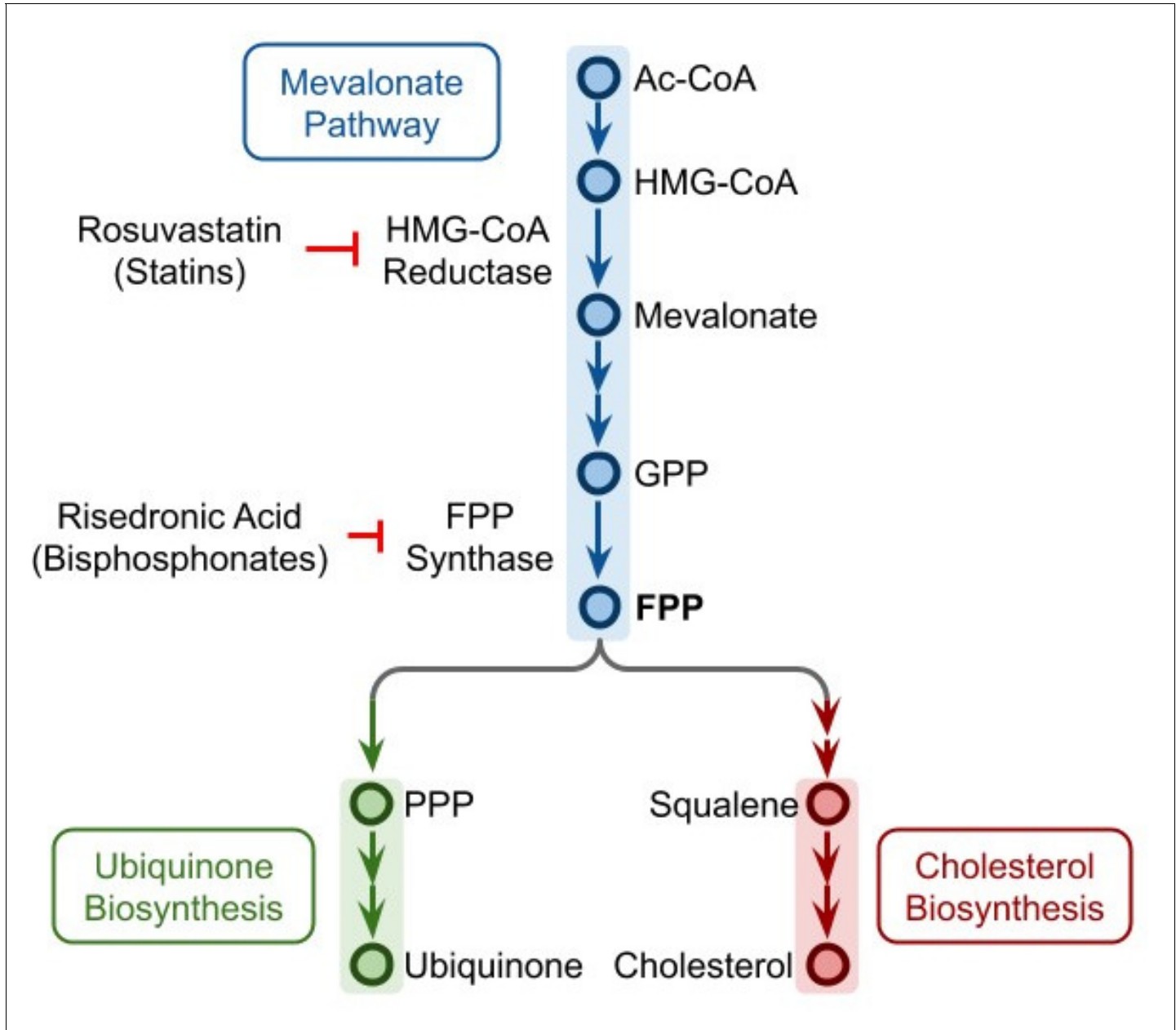

**Figure 2.** Ubiquinone and cholesterol biosynthesis pathway. Ubiquinone and cholesterol biosynthesis pathways originate from a branching of the mevalonate pathway at FPP. Rosuvastatin and other statins can inhibit the HMG-CoA reductase, while risedronic acid and other bisphosphonates can inhibit the FPP synthase. Ac-CoA: acetyl coenzyme A, HMG-CoA: hydroxymethylglutaryl coenzyme A, GPP: geranyl pyrophosphate, FPP: farnesyl pyrophosphate, PPP: polyprenyl pyrophosphate.

Several ophthalmic items also appear to be associated with significantly decreased odds for hospitalization, including artificial tears, hydroxypropyl-methylcellulose-based (OR=0.657, CI 0.490 to 0.871), or hyaluronic acid based (OR=0.322, CI 0.098 to 0.836); decreased OR are also found for items that may act as a physical barrier to the eye: eye care wipes, which are sterile wipes sold to clean the eyes (OR=0.285, CI 0.054 to 0.956), a retinol-based ointment used to treat cornea abrasion (OR=0.285, CI 0.054 to 0.956), and timolol drops used to treat glaucoma (OR=0.570, CI 0.328 to 0.949).

Also associated with decreased odds for hospitalization are several drugs based on an ACE inhibitor or an angiotensin receptor blocker (ARB), sometimes in combination with another compound. In addition to cilazapril and ramipril-hydrochlorothiazide that were highly ranked in both cohorts, *cohort 1* identifies candesartan (OR=0.718, CI 0.544 to 0.934), and *cohort 2* identifies valsartan with amlodipine (OR=0.821, CI 0.698 to 0.963), and losartan with hydrochlorothiazide (OR=0.783, CI 0.630 to 0.969).

Remarkably, several drugs acting on receptors to neurotransmitters also appear to decrease hospitalization risk: rizatriptan (OR=0.118, CI 0.003 to 0.750), bupropion (OR=0.399, CI 0.136 to 0.976), and methylphenidate (OR=0.444, CI 0.178 to 0.972).

In the Israeli population, the two groups that have been reported to be at higher risk are Ultra-Orthodox Jews and Arabs (*Muhsen et al., 2021*). Therefore, we performed additional analyses with the goal to eliminate membership in either of these groups as a potential confounder and to eliminate possible confounding in concurrently used medications. We performed multivariate conditional logistic regression (Materials and methods) in each of the cohorts. In each cohort, we modelized the OR for hospitalization, using ethnicity and purchased medications as explanatory factors. See *Supplementary file 1*. Either Ultra-Orthodox or Arab identity indeed appear to be each associated with increased risk for hospitalization. However, even after adjusting for the subgroup membership, most of the medications identified by individual Fisher's exact tests maintain statistically significant protective effect.

Because of the established association between high BMI and COVID-19 severity, it is of interest to know whether any of the protective medications are especially protective in high BMI individuals. Therefore, we performed a subgroup analysis, by partitioning partition BMI into four ranges (see Materials and methods). The results are shown as a forest plot in *Supplementary file 1*-table 3. In general, the protective effects were seen in most or all BMI ranges and we did not see any striking association between a protective medication and high BMI.

## Discussion

In this large-scale retrospective study, we identified several drugs and products that are significantly associated with reduced odds for COVID-19 hospitalization, both in the general population and in patients with laboratory-proven SARS-CoV-2 infection. Several other research groups have recognized the potential for EHRs to enable large-scale studies in COVID-19 and the challenges of this sort of retrospective research are reviewed in *Dagliati et al., 2021*; *Sudat et al., 2021*. To give a few examples, EHRs have also been used to predict: (i) COVID-19 mortality based on pre-existing conditions (*Estiri et al., 2021*; *Osborne et al., 2020*), (ii) early diagnosis of COVID-19 based on clinical notes (*Wagner et al., 2020*), and (iii) eligibility of COVID-19 patients for clinical trials by matching trial criteria with patient records (*Kim et al., 2021*).

Major strengths of our study include: (i) the large sample of hospitalized COVID-19 patients, (ii) the ability to collect comprehensive data about individual demographic and comorbidity characteristics and to build matched case and control populations, (iii) the ability to track hospitalizations and disease severity, owing to a central database established by the Israeli Ministry of Health, and (iv) the capacity to track which drugs and products have been acquired by patients in the period that have preceded SARS-CoV-2 infection, owing to comprehensive digital systems integration in CHS.

Another strength is the dual cohort design, with control individuals taken from the general population in the first cohort and from individuals positive for SARS-CoV-2 in the second cohort, with each using different matching criteria, mitigates potential bias that could affect each cohort. The two cohorts allowed us to evaluate the protective effect of drugs that act either by reducing the initial risk of infection or by reducing the risk of hospitalization in those infected. Analyses are based

on items procured in the 35 days before the initial positive test. This window was chosen in accordance with the monthly renewal of prescription policy in place in CHS.

Limitations of this study are related to it being observational in nature. Best efforts were made to use matching so that patients in case and controls are similar regarding most of the known factors for disease severity, and notably, age, obesity, smoking, and baseline comorbidity. The cases and controls were not matched for ethnicity, which could be a substantial confounding factor. We aimed to get a sensible tradeoff between controlling for confounding factors by rigorous matching and keeping enough patients so that cohorts are representative of the general population. Our analysis is based on medication acquisition in pharmacies and does not ascertain that medications purchased were used. Notably, some of the drugs associated with a protective effect may have been stopped during patient's hospitalization so that our analysis may have underestimated the full achievable benefits for some of the drugs. Conversely, since drugs tested here were acquired before patients were positive for SARS-CoV-2, the protective effect of some of the drugs may be fully attained only when treatment is started before or early in the infection.

The variable behavior of people during the pandemic has been an important factor that can affect the risk of exposure and the severity of infection. We tried to address this cause of variable risk by performing matching in two distinct cohorts and by using only PCR-positive patients in the second cohort. Nevertheless, behavioral factors, which could not measure, can still account for some of the observed differences.

Our analyses counted the purchase of each medication, but not the dose or the patient compliance. Therefore, we cannot comment on whether higher doses of the beneficial medications, such as rosuvastatin and ubiquinone, are associated with reduced risk.

The medications that are protective are prescribed for a variety of conditions. It is conceivable but unlikely that it is the medical condition, or comorbidity, that provides the protection rather than the medication itself. Three of the comorbidities that have been prominently suggested as relevant to COVID-19 severity and outcome include high BMI, diabetes, and hypertension. Therefore, at the helpful suggestion of the reviewers, we did both subgroup analysis and regression analysis to show that the protective effect of the most protective medications appears not to be associated with BMI (*Supplementary file 1*). The study design explicitly matched for diabetes and hypertension, so it follows that these two diseases are not associated with the protective effects of the drugs listed in *Table 2A and B*. However, we recognize the limitation that when the association between the medical condition and the prescription is very specific, such as flecainide for cardiac arrhythmia, we lack suitable data to separate the possible effects of the condition and the medication.

Bearing these strengths and potential limitations in mind, our analyses seem to indicate several viral vulnerability points, which can potentially be exploited to effectively reduce disease severity with drugs that are already available. The drugs identified as protective include ubiquinone, which is a food supplement with a very good safety profile that does not even require a prescription in our health system, and rosuvastatin and ezetimibe, two drugs prescribed routinely to reduce cholesterol and that have a very good safety profile. These findings are in line with previous reports that RNA viruses need cholesterol to enter cells, for virion assembly, and to maintain structural stability (*Aizaki et al., 2008*; *Bajimaya et al., 2017*; *Rossman et al., 2010*; *Sun and Whittaker, 2003*), and that prescribing statins may protect against infection with RNA viruses such as members of family Flaviviridae, including dengue virus, Zika virus, and West Nile virus (*Gower and Graham, 2001*; *Osuna-Ramos et al., 2018*; *Whitehorn et al., 2016*). The involvement of the cholesterol/ubiquinone pathway is further confirmed by the fact that risedronic acid, a drug acting on the enzyme farnesyl pyrophoshate synthase (*Tsoumpra et al., 2015*; *Figure 2*) which catalyzes the production of FPP from which the cholesterol and the ubiquinone synthesis pathways split (*Buhaescu and Izzedine, 2007*), is identified as protective as well, even though it is prescribed for osteoporosis regardless of the presence of hypercholesterolemia.

Taken together, our findings lend (albeit indirect) support to the possibility that SARS-CoV-2 hijacks the cholesterol synthesis pathway, possibly to boost production of the cellular cholesterol it needs as an RNA virus. The fact that ubiquinone protects against severe disease suggests that SARS-CoV-2 may tilt the mevalonate pathway toward cholesterol synthesis and away from ubiquinone synthesis. Such a pathway imbalance would ultimately result in deficiency of ubiquinone that could lead to cell death unless counteracted by ubiquinone supplementation.

It is remarkable that the protective effect of anti-cholesterol drugs was observed mostly for rosuvastatin – and in *cohort 1* for pravastatin – but not for other statins. Rosuvastatin was found to significantly increase 25-OH vitamin D levels in the blood (*Yavuz et al., 2009*), much more than what could be observed with other statins. *Yavuz and Ertugrul, 2012* suggested that the increase in 25-OH vitamin D observed following rosuvastatin treatment could be mediated by the Niemann-Pick C1 like 1 (NPC1L1) membrane transporter that is involved in intestinal absorption of vitamin D. Interestingly, the NPC1L1 membrane transporter is also the target of ezetimibe, identified by our study to decrease significantly the hospitalization risk of COVID-19 patients.

In both cohorts, we observed a significant decrease of the odds for hospitalization for COVID-19 patients treated with either vitamin D or magnesium citrate. Vitamin D deficiency has been shown to be associated with increased risk for COVID-19 in multiple studies (*Israel et al., 2020*; *Merzon et al., 2020*). Magnesium is needed for vitamin D activation (*Uwitonze and Razzaque, 2018*) and its levels in drinking water in Israel are low, as water is produced in great part through desalination of sea water (*Koren et al., 2017*). The decreased hospitalization rate revealed here for patients taking magnesium supplementation may suggest a role for supplementation of this element along with vitamin D. Hospitalization risk was also found to be decreased in patients taking vitamin B12 and calcium-zinc, as identified by other studies (*Ragan et al., 2020*; *Trasino, 2020*; *Wessels et al., 2020*).

Another medication that was associated with decreased odds for hospitalization is flecainide, an antiarrhythmic drug that blocks sodium channels in the heart, and inhibits ryanodine receptor 2, a major regulator of sarcoplasmic release of stored calcium ions. It may prevent apoptosis by release of calcium from the endoplasmic reticulum (ER) once the cell mitochondria cease to function. An expert review recommended that patients with arrhythmia who get COVID-19 should continue flecainide treatment if already prescribed (*Ci et al., 2020*). In our study, the protective effect observed in both cohorts is even more marked for severe patients, suggesting that this drug, which can be given intravenously (*Antonelli et al., 2006*), could be administered to patients in respiratory distress, if the protective effect is confirmed in clinical trials.

Several drugs acting as ACE inhibitors or ARBs appeared to slightly decrease the odds for hospitalization, either alone or in combination (cilazapril, ramipril-hydrochlorothiazide, losartan-hydrochlorothiazide, valsartan amlodipine). These results are consistent with ACE and ARBs treated patients shown to not have an increased risk for COVID-19 (*Morales et al., 2021*), and there is therefore no reason to discontinue these medications to decrease COVID-19 risk. Our findings also confirm and substantially extend recent EHR-based findings about the favorable association between metformin use and COVID-19 outcomes (*Bramante et al., 2021*).

In addition, several drugs acting on synapses (escitalopram, bupropion, mirabegron, and timolol) were associated with decreased risk of hospitalization. This is consistent with SARS-CoV-2 invading neuronal cells (*Iroegbu et al., 2020*; *Meinhardt et al., 2021*; *Song et al., 2021*), as manifest by symptoms of loss of smell and taste, where it may spread throughout the nervous system across synapses. Decreased neurotransmitter internalization may therefore reduce the infectious potential of the virus.

Interestingly, items that could improve the physical barrier of the eye surface were among the top items decreasing odds of hospitalization, including eye wipes, artificial tears, and eye ointments. Interestingly, the protective effect for these items was observed foremost among patients from *cohort 2* in which controls are already SARS-CoV-2 positive. This suggests that these barrier items could not only protect against the initial risk of infection, but, notably, also reduce disease severity in patients already infected. The beneficial effect observed here for many different ophthalmologic preparations raises the possibility that autoinoculation of the virus to the eyes, prevented by these items, has a role in the virulence of SARS-CoV-2. The possibility that invasion of the central nervous system by the virus through the eyes could increase the risk of COVID-19 complications is also supported by the fact that eyeglass wearers were shown previously to be at decreased risk for COVID-19 hospitalization (*Zeng et al., 2020*). Until the meaning of these findings is fully understood, it may be helpful to advise COVID-19 patients to avoid touching their eyes in order to reduce the risk of complications.

In conclusion, this study shows apparently protective effects for several medications and dietary supplements, such as rosuvastatin, ezetimibe, ubiquinone, risedronate, vitamin D, and magnesium. We suggest to further investigate these, and other products identified by this study, in prospective

trials aimed to reduce disease severity in COVID-19 patients. In the meantime, we believe that the observed protective effects of these drugs provide important evidence supporting their safe continuation for COVID-19 patients.

## Acknowledgements

The content of this publication does not necessarily reflect the views or policies of the Department of Health and Human Services, nor does mention of trade names, commercial products, or organizations imply endorsement by the US Government.

## Additional information

### Funding

| Funder | Grant reference number | Author |
| --- | --- | --- |
| National Cancer Institute | Intramural funding | Alejandro A Schäffer<br>Kuoyuan Cheng<br>Sanju Sinha<br>Eytan Ruppin |

The funders had no role in study design, data collection and interpretation, or the decision to submit the work for publication.

### Author contributions

Ariel Israel, Conceptualization, Data curation, Software, Investigation, Methodology, Writing - original draft, Writing - review and editing; Alejandro A Schäffer, Conceptualization, Writing - original draft, Writing - review and editing; Assi Cicurel, Conceptualization, Investigation, Writing - original draft, Writing - review and editing; Kuoyuan Cheng, Sanju Sinha, Eyal Schiff, Conceptualization, Investigation, Writing - review and editing; Ilan Feldhamer, Validation, Investigation, Writing - review and editing; Ameer Tal, Software, Validation, Investigation, Writing - review and editing; Gil Lavie, Eytan Ruppin, Conceptualization, Supervision, Investigation, Methodology, Writing - original draft, Writing - review and editing

### Author ORCIDs

Ariel Israel ![ORCID] https://orcid.org/0000-0003-4389-8896
Alejandro A Schäffer ![ORCID] https://orcid.org/0000-0002-2147-8033
Eytan Ruppin ![ORCID] https://orcid.org/0000-0002-7862-3940

### Ethics

Human subjects: This study has been approved by the CHS Institutional Review Board (IRB) with a waiver of informed consent, approval number: COM-0046-20.

### Decision letter and Author response

Decision letter https://doi.org/10.7554/eLife.68165.sa1
Author response https://doi.org/10.7554/eLife.68165.sa2

## Additional files

### Supplementary files

• Source code 1. R source code, producing *Figure 1*.

• Supplementary file 1. Supplementary tables and figures. Suppl Tab 1 Multivariable logistic regression for hospitalization status according to ethnicity and medication consumption in Cohort 1. Suppl Tab 2 Multivariable logistic regression for hospitalization status according to ethnicity and medication consumption in Cohort 2. Figure supplements Forest plot showing association between drug use and hospitalization risk in each of the cohorts, divided by body mass index (BMI) category.

• Transparent reporting form

### Data availability

Data were obtained from patients' electronic health records, and IRB approval restrains its use to researchers inside Clalit Health Services. For further information regarding data availability, researchers may contact Dr. Lavie gillav@clalit.org.il. This study is based on real-world patient drug purchases, and it cannot be made available due to patient privacy concerns. R code used to produce Figure 1 is available as a supplementary file named 'Source code 1'.

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
