## [Decision Letter]

**Acceptance summary:**

Your work is of interest to epidemiologists working on COVID19 research or clinical scientists with an interest in prospective trials to study drugs to reduce COVID19 hospitalization. It provides an extensive list of drugs that are associated with reduced hospitalization for COVID19, studied in both in a patient-control cohort as well as positive vs hospitalized SARS-CoV-2 patients. The findings that cholesterol lowering drugs and drugs to treat the eyes are novel and of interest and might open and support explorative trials with these drugs in COVID19.

**Decision letter after peer review:**

Thank you for submitting your article "Identification of drugs associated with reduced severity of COVID-19: A case-control study in a large population" for consideration by *eLife*. Your article has been reviewed by 2 peer reviewers, including Frank L van de Veerdonk as the Reviewing Editor and Reviewer #1, and the evaluation has been overseen by Jos van der Meer as the Senior Editor.

Essential revisions:

1) Both reviewers felt that there are some limitations to the study that have not been addressed. Importantly the behavior of people during the pandemic can be a huge factor of exposure and risk of severity of infection. Although the authors address this by using the second cohort, which is also PCR positive patients, it could still influence the load of exposure, and thus severity of disease. This discussion on behavior and possible outcome needs to be included in the manuscript.

2) What if not the drugs, but the underlying disease or other comorbidities are the protective factor? This could be further analyzed using different statistical methods such as linear regression models instead of Fishers exact test.

3) During the matching, did you also match for etnicity? This could be a strong confounder and might be necessary to be taken into account

4) The matching for co-morbidities is indeed taken into account. However, the severity of the condition which can be reflected by the use and dose of medication is not taken into account. Understandable since this would need a significant amount of extra information and statistics, but at least it needs to be discussed.

5) This might be phrasing, but did you match for gender or sex? These are two different things

6) Could you elaborate and perhaps perform extra analyses on the effects of the underlying comorbidities and BMI on the findings? Were the findings stronger or weaker in some subsets of patients / weight? Did they correlate with the diseases that the medication is used for? This might help unravel the question whether the drug or the disease causes this effect. Instead of using Fishers test, you could use linear regression models to analyze this.

---

## [Author Response]

Essential revisions:1) Both reviewers felt that there are some limitations to the study that have not been addressed. Importantly the behavior of people during the pandemic can be a huge factor of exposure and risk of severity of infection. Although the authors address this by using the second cohort, which is also PCR positive patients, it could still influence the load of exposure, and thus severity of disease. This discussion on behavior and possible outcome needs to be included in the manuscript.

We thank the reviewers for pointing out this important caveat. Accordingly, we added the following text in the Discussion:

“The variable behavior of people during the pandemic is an important factor that can affect the risk of exposure, and the severity of infection. […] Nevertheless, behavioral factors, which we could not measure, can still account for some of the observed differences.”

“Our analyses counted the purchase of each medication, but not the dose or the patient compliance. Therefore, we cannot comment on whether higher doses of the beneficial medications, such as rosuvastatin and ubiquinone, are associated with reduced risk.”

2) What if not the drugs, but the underlying disease or other comorbidities are the protective factor? This could be further analyzed using different statistical methods such as linear regression models instead of Fishers exact test.

This is an interesting possibility in some cases, but implausible in others. For example, it seems implausible that poor vision for which eyeglasses are prescribed would be protective. And even if it were, we see no way to separate the effect of the poor vision from the effect wearing of eyeglasses to correct the poor vision. For some comorbidities, the comment overlaps with comment 6. We added to the Discussion the following text:

“The medications that are protective are prescribed for a variety of conditions. […] However, we recognize the limitation that when the association between the medical condition and the prescription is very specific, such as flecainide for cardiac arrhythmia, we lack suitable data to separate the possible effects of the condition and the medication.”

See also the response to point 6.

3) During the matching, did you also match for etnicity? This could be a strong confounder and might be necessary to be taken into account

This is an excellent point, thanks. The population was not matched for ethnicity in the initial analysis, but this issue was addressed indirectly and to some extent by matching for socio-economic status, which was mentioned in the original submission.

In the Israeli population the two groups that have been reported to be at higher risk for covid-19 are Ultra-Orthodox Jews and Arabs (K. Muhsen et al. The Lancet Regional Health Europe 2021; 7:100130, PMID 34109321). Therefore, in response to your comment we now performed additional analyses aiming to eliminate membership in either of these groups as a potential confounder, and also to rule out their possible confounding effect in the analysis of concurrently used medications. We performed multivariable conditional logistic regression analyses in each of the cohorts. In each cohort, we modeled the odds ratio for hospitalization, using ethnicity and purchased medications as explanatory factors. See Supplementary Tables 1 and 2 in Additional file 1.

In these additional regression analyses, either Ultra-Orthodox or Arab identity indeed appear to be each associated with increased risk for hospitalization compared with the reference general population, consistent with recent reports and other studies (Muhsen et al., 2021). However, even after adjusting for the subgroup membership, most of the medications identified by individual Fisher tests maintain statistically significant protective effect.

We added text similar to the above explanation to Methods and Results.

4) The matching for co-morbidities is indeed taken into account. However, the severity of the condition which can be reflected by the use and dose of medication is not taken into account. Understandable since this would need a significant amount of extra information and statistics, but at least it needs to be discussed.

Indeed, underlying diseases and comorbidities may also affect risks. Our study used cohorts matched for main known comorbidity risks, such as obesity (including body mass index categories), hypertension, diabetes, asthma, COPD, ischemic heart disease, congestive heart failure, renal failure, and malignancy to account for these factors. Nevertheless, disease severity, and different patterns of use and dose of medication may have also affected the risk, in a manner difficult to assess using the chosen methodology. We hence now added in the Discussion under the limitations:

“Our analyses counted the purchase of each medication, but not the dose or the patient compliance. Therefore, we cannot comment on whether higher doses of the beneficial medications, such as rosuvastatin and ubiquinone, are associated with reduced risk.”

5) This might be phrasing, but did you match for gender or sex? These are two different things

We thank the reviewers for pointing out this error. The matching was performed for sex. We changed the three occurrences of “gender” in the original submission to “sex” in the revised submission.

6) Could you elaborate and perhaps perform extra analyses on the effects of the underlying comorbidities and BMI on the findings? Were the findings stronger or weaker in some subsets of patients / weight? Did they correlate with the diseases that the medication is used for? This might help unravel the question whether the drug or the disease causes this effect. Instead of using Fishers test, you could use linear regression models to analyze this.

Both our cohorts were matched for BMI, diabetes and hypertension status, which are the main known factors for disease severity, so these are already accounted for in our analyses. Since these variables are already matched for in the cohorts, we cannot use these factors as variables in regression models, as suggested by the reviewer. However, we can elucidate whether the effect was stronger or weaker for some patients' weight, by performing subgroup analyses (i.e. each of the cohorts can be subdivided by BMI categories (<25, 25-30, 30-35, >35), and the statistical analyses performed separately in each of the subgroups). Following the reviewers' remark, we performed such a subgroup analysis, and display the results as a forest plot, in the newly added Additional File 1. This analysis shows that for most of the identified drugs, reduced risk is observed in each of the BMI subgroups – even though the smaller size of some subgroups does not always allow to reach statistical significance.

We added the following text to Methods:

“To assess for possible associations between the protective effect of a medication and BMI, we partition the matched subjects into four BMI ranges: <25, 25-30, 30-35, >35. Then we redid our association analyses in each range.”

We added the following text to Results:

“Because of the established association between high BMI and COVID-19 severity, it is of interest to know whether any of the protective medications are especially protective in high BMI individuals. […] In general, the protective effects were seen in most or all BMI ranges and we did not see any striking association between a protective medication and high BMI.”